# MicroRNA, Diabetes Mellitus and Colorectal Cancer

**DOI:** 10.3390/biomedicines8120530

**Published:** 2020-11-24

**Authors:** Hsiuying Wang

**Affiliations:** Institute of Statistics, National Chiao Tung University, Hsinchu 30010, Taiwan; wang@stat.nctu.edu.tw

**Keywords:** biomarkers, colorectal cancer, diabetes mellitus, glucose, insulin, microRNA

## Abstract

Diabetes mellitus (DM) is an endocrinological disorder that is due to either the pancreas not producing enough insulin, or the body does not respond appropriately to insulin. There are many complications of DM such as retinopathy, nephropathy, and peripheral neuropathy. In addition to these complications, DM was reported to be associated with different cancers. In this review, we discuss the association between DM and colorectal cancer (CRC). CRC is the third most commonly diagnosed cancer worldwide that mostly affects older people, however, its incidence and mortality are rising among young people. We discuss the relationship between DM and CRC based on their common microRNA (miRNA) biomarkers. miRNAs are non-coding RNAs playing important functions in cell differentiation, development, regulation of cell cycle, and apoptosis. miRNAs can inhibit cell proliferation and induce apoptosis in CRC cells. miRNAs also can improve glucose tolerance and insulin sensitivity. Therefore, investigating the common miRNA biomarkers of both DM and CRC can shed a light on how these two diseases are correlated and more understanding of the link between these two diseases can help the prevention of both DM and CRC.

## 1. Introduction

Diabetes mellitus (DM), referred to simply as diabetes, is a metabolic disease that is due to either the pancreas not producing enough insulin or the cells of the body not responding properly to insulin. Diabetes is caused by hyperglycemia, and chronic hyperglycemia is associated with long-term damage and dysfunction of different organs such as the eyes, kidneys, nerves, heart, and blood vessels [1]. According to the National Diabetes Statistics Report in 2020, 34.2 million people have diabetes, 88 million people aged 18 years or older have prediabetes, and 24.2 million people aged 65 years or older have prediabetes in the United State [2].

There are three major DM types: (1) type I DM: the pancreas fails to produce insulin; (2) type 2 DM: the body does not respond appropriately to insulin; (3) gestational DM: this type occurs in pregnant women when the body becomes less sensitive to insulin. Gestational DM is a disease with onset or first recognition during pregnancy in women without previously diagnosed DM [3]. In type 1 DM, individuals’ immune systems attack the insulin-producing β-cells in the pancreas, and this can cause the pancreas to stop generating enough insulin to maintain normal levels of glucose in the blood. Type 1 DM without treatment may lead to serious health complications. In type 2 DM, individuals have insulin resistance. These individuals do not need insulin treatment to survive. Most patients with this form of DM are obese, and obesity may cause some degree of insulin resistance [1].

There are many complications of DM such as retinopathy, nephropathy, peripheral neuropathy, autonomic neuropathy, cardiovascular symptoms, and sexual dysfunction. In addition, patients with DM have an increased incidence of hypertension, atherosclerotic cardiovascular, peripheral arterial, and abnormalities of lipoprotein metabolism. Diabetic retinopathy (DR) is a common neurovascular complication of DM. Pregnancy increases the short-term risk of DR that is a leading cause of blindness in pregnant women [4]. Current therapeutics of DR target retinal edema and neovascular lesions. However, neurodegeneration may contribute to the development of microvascular dysfunction and neovascularization. Several studies have demonstrated that fenofibrate, a PPARα agonist used to treat dyslipidemia, had unprecedented therapeutic effects in DR [5]. Diabetic kidney disease (DKD) or diabetic nephropathy is a type of chronic kidney disease caused by DM. DKD is the leading cause of chronic kidney disease and has been reported in approximately 40% of DM patients. The majority of DKD patients die from cardiovascular diseases and infections before needing kidney replacement therapy [6].

In addition to the above-mentioned complications, DM was reported to be associated with different cancers. Epidemiologic evidence suggests that people with DM are at significantly higher risk for many cancers, including liver, pancreas, endometrium, colon and rectum, breast, and bladder cancers [7]. In this review, we discuss the association between DM and colorectal cancer (CRC). CRC that starts in the colon or the rectum is the third most commonly diagnosed cancer worldwide. CRC mostly affects older people. However, its incidence and mortality are rising among young people [8]. Lately, the death rate from CRC has decreased because of the progress in screening techniques and improvements in treatments. CRC patients in its earliest stage usually have surgery as the first treatment, and chemotherapy may be used after surgery. For metastatic CRC patients, however, surgery and chemotherapy are not satisfactory treatments. Instead, targeted therapy that is a new option has successfully prolonged overall survival for CRC patients [9]. In addition, chemotherapy includes fluoropyrimidine (5-FU)-based therapy, which is the gold standard of first-line treatment for CRC.

There have been many studies investigating the relationship between DM and CRC based on clinical cohort studies. A meta-analysis based on the cohort studies found that colorectal, colon, and rectal cancer patients with DM had a 5-year shorter survival compared to patients without DM, with an 18%, 19%, and 16% decrease in overall survival respectively [10]. CRC patients with DM are at greater risk for all-cause and cancer-specific mortality and have worse disease-free survival compared to those without DM [11]. Among colon cancer patients with DM who receive antidiabetic drug therapy, patients who use insulin have shorter overall survival and cancer-specific survival than patients who do not [12]. Patients with DM and high-risk stage II and stage III colon cancer experienced a significantly higher rate of overall mortality and cancer recurrence [13]. In addition, an optimal glycemic control level was recommended as an HbA1c of 7.8% or below for colon cancer patients with DM [14]. In a study, 520 CRC patients were classified into two groups according to their blood sugar levels (≧ 110 or <110 mg/dL). In addition to these cohort studies, in this review, we discuss the relationship between DM and CRC based on their biological biomarkers.

## 2. MicroRNA

MicroRNAs (miRNAs), about 21–24 nucleotides in length, are non-coding RNAs playing important functions in cell differentiation, development, regulation of cell cycle, and apoptosis [15]. They also act either as tumor suppressors or oncogenes and play a role in tumorigenesis by regulating some oncogenes and tumor suppressor genes [16]. On the contrary, miRNAs are also regulated by tumor suppressor genes and oncogenes [17]. The first miRNA was discovered in the 1990s while studying the nematode Caenorhabditis elegans regarding the gene lin-14 [18]. miRNA can regulate mRNA by binding to 3′-untranslated regions, and it has been estimated that miRNAs may regulate up to 30% of the protein-coding genes in the human genome [15,19].

The biogenesis of miRNA is categorized into canonical and non-canonical pathways. In the canonical biogenesis pathway, a primary miRNA transcript (pri-miRNA) is cleaved by the endoRNase Drosha to excise the precursor miRNA (pre-miRNA). The nuclear RNase III Drosha cleaves pri-miRNAs to release pre-miRNAs that are cut by the cytoplasmic RNase III Dicer to process into mature miRNAs [20,21]. Many non-canonical miRNA biogenesis pathways have been elucidated, that use different combinations of the proteins involved in the canonical pathway. In general, the non-canonical pathways are grouped into Drosha/DGCR8-independent and Dicer-independent pathways [21].

miRNAs are especially involved in the initiation and progression of cancers and are useful biomarkers for different cancers [22,23,24,25,26,27,28,29,30]. In addition to cancer, miRNAs also contribute to other diseases including neurological diseases and inflammation such as amyotrophic lateral sclerosis [31,32], Parkinson’s disease [32,33], and anti-NMDA receptor encephalitis [34,35,36]. Additionally, miRNA can be used to explore the association between diseases or between disease and vaccination [37,38,39]. There are many miRNA biomarkers discovered for DM and CRC, respectively. miRNAs are also discovered to be associated with both DM and CRC. Obesity is associated with insulin resistance that is a risk factor for CRC, and miRNAs are found to be similarly dysregulated in obesity, insulin resistance, and CRC [40]. In the next section, we discuss their common miRNAs.

## 3. MicroRNA Biomarkers

The association between DM and CRC based on their miRNA biomarkers are discussed in this section. I collected miRNA biomarkers of DM and CRC from the literature, respectively, and then found the common miRNA biomarkers for both diseases. Since there are many common miRNA biomarkers for both diseases, in this review, I discuss some of these common miRNA biomarkers and list them in Table 1.

Restoration of miR-92a levels in CD34(+) cells from DM patients with diabetic retinopathy reduced the inflammatory phenotype of these cells which suggested that restoring levels of miR-92a could enhance the usefulness of CD34(+) cells in autologous cell therapy [41]. Overexpression of miR-92a reduced insulin expression while inhibition of miR-92a expression promoted insulin expression and ultimately enhanced glucose-induced insulin secretion [42]. miR-92a targeted the anti-apoptotic molecule BCL-2-interacting mediator of cell death in colon cancer tissues [43]. The serum expression level of miR-766 was lower in patients with type 2 DM, compared with those of healthy subjects [44]. Structured exercises are of great benefit for type 2 DM patients. Compared with baseline, the post-training level of miR-766-3p was significantly up-regulated in a study of 24 selected patients randomized to aerobic or resistance training protocols [45]. The overexpression of miR-766 reduced cell growth in colon cancer cells through suppression of the MDM4/p53 pathway. On the contrary, the downregulation of miR-766 promoted cell growth in colon cancer cells [46]. 5-Fluorouracil (5-FU) is a classic chemotherapeutic drug for CRC treatment. miR-96 may modulate 5-FU sensitivity in CRC cells [47]. miR-96 promotes the pathogenesis of hepatic insulin resistance resulted from saturated fatty acids or obesity [48]. The up-regulation of miR-96 contributes to the development of insulin resistance by targeting insulin receptor substrate 1 (IRS-1) in SK-Hep1 cells [49]. Circulating miR-100 was significantly lower in obese normoglycemic subjects and subjects with type 2 DM [50]. miR-100-5p was dysregulated in type 1 DM patients compared to controls [51]. miR-100 plays a tumor suppressor role by regulating CRC cell growth and invasion phenotype [52].

Restoration of miR-365 expression inhibited cell cycle progression and promoted 5-FU-induced apoptosis in colon cancer cell lines [53]. The inhibition of miR-365 led to an increase of Timp3. Both miR-365 and Timp3 may represent potential therapeutic targets for the treatment of DR [54]. miR-365a-3p was showed to have a very significant correlation with hyperglycemia levels [55]. The increased miR-365 in glyoxal-treated rat Müller cell line is involved in DR through the miR-365/Timp3 pathway and oxidative stress mechanism [56]. The miR-378 expression profile was significantly higher in type 1 DM patients compared with the controls [57]. miR-378-5p down-regulated BRAF in CRC cells [34]. In an analysis of associations between miRNA expression and titers of islet autoantibodies (GADA, IA2A, IAA and the three variants of ZnT8A: Trp/Arg/Glt), GADA titers correlated positively to miR-378a-3p, IA2A correlated negatively to miR-378a-3p and ZnT8A (Trp) correlated negatively to miR-378a-3p [58]. CRC patients with a high tumor miR-18a level tend to have a quicker recurrence after surgery, compared to patients with a low tumor miR-18a level [59]. miR-18a in peripheral blood mononuclear cells may be an important marker of stress reaction and may play a role in vulnerability to type 2 DM as well as insulin resistance. The degrees of insulin resistance was measured using the homeostasis model assessment of insulin resistance (HOMA-IR) in a study of three groups of study subjects were involved, including type 2 DM patients, impaired fasting glucose (IFG) individuals, and healthy controls. The increased levels of miR-18a were associated with the risk of type 2 DM and IFG and miR-18a was an independent positive predictor of HOMA-IR [60]. miR-18a-3p was associated with hemoglobin A1c (HbA1c) levels [61]. Overexpression of miR-125a-5p inhibited cell proliferation and induced apoptosis in colon cancer cells [62]. The miR-125a-5p level is decreased in livers of type 2 DM rats and mice [63]. miR-125a is over-expressed in insulin target tissues in a spontaneous rat model of type 2 DM [64]. miR-125b may promote apoptosis in CRC cell lines by suppressing the anti-apoptotic molecules of the BCL-2 family [65]. Up-regulated miR-125b-5p promotes insulin sensitivity and enhances pancreatic β-cell function through inhibiting the JNK signaling pathway by negatively mediating DACT1 [66]. The expression level of miR-125b was elevated in peripheral blood mononuclear cell samples from patients with type 2 DM [67]. High levels of miR-125b are associated with HbA1c in prediabetic, type 2 DM, and type 1 DM [68]. miR-200c functions as an oncogene in colon cancer cells by regulating tumor cell apoptosis, survival, invasion, and metastasis [69]. miR-200c is a mediator of diabetic endothelial dysfunction and inhibition of miR-200c rescues endothelium-dependent relaxations in diabetic mice [70]. miR-200c-3p was positively correlated with HbA1c [55].

miR-206 may inhibit cell proliferation by arresting the colon cancer tumor cells at the G1/G0 phase and accelerating apoptosis [71]. The upregulation of miR-206 inhibited cancer cell proliferation and activated apoptosis in colon cell lines by targeting NOTCH3 [72]. miRNA-206 is a potent inhibitor of lipid and glucose production by simultaneously facilitating insulin signaling and impairing hepatic lipogenesis [73].

Glucokinase (GK) is rate-limiting for glucose-stimulated insulin secretion (GSIS) from pancreatic islets. The loss of miR-206 increases GK activity in islets and liver, leading to improved glucose tolerance and GSIS [74]. Overexpression of miR-210 induces apoptosis in CRC that is associated with an upregulation of pro-apoptotic Bim expression and Caspase 2 processing [75]. miR-210 derived from adipose tissue macrophages promotes mouse obese diabetes pathogenesis by regulating glucose uptake and mitochondrial complex IV activity [76]. Downregulation of miR-23a promoted cell apoptosis in microsatellite instability (MSI) CRC cells treated with 5-FU. miR-23a, targeting ABCF1, enhances 5-FU resistance in MSI CRC cells [77]. The results of the qRT-PCR assessment showed that the levels of miR-23a significantly declined in type 2 DM patients compared with pre-diabetes patients [78]. miR-23a-3p was decreased in impaired glucose tolerance compared to normal glucose tolerance [79].

miR-129-5p was significantly overexpressed in Langerhans islets transplantation patients [80]. The expression of miR-129 was significantly downregulated in CRC tissue specimens compared with the paired normal control samples [81]. Glucose up-regulated miR-218 expression, and miR-218 could inhibit the proliferation and facilitate the apoptosis of human RPE cells by targeting runt-related transcription factor 2 [82]. Syntaxin-binding protein 1 (Stxbp1) plays an essential role in exocytosis and is crucial for insulin secretion. Stxbp1 was downregulated by miR-218 [83]. miR-218 was revealed to inhibit adiponectin-induced AMP-activated protein kinase (AMPK) and p38 mitogen-activated protein kinase (MAPK) activation and glucose uptake in HepG2 cells [84]. There is a higher urinary exosomal miR-218 expression in type 1 DM in children than in healthy controls [85]. miR-218 was decreased while c-FLIP expression was elevated in human colon cancer tissues [86]. miR-195-5p expression was significantly increased in serum samples from gestational DM patients as compared with that in healthy pregnancies [3]. miR-195 regulates sirtuin 1 (SIRT1)-mediated tissue damage in diabetic retinopathy [87]. Knockdown of miR-195 increased myocardial capillary density and improved maximal coronary blood flow in diabetic mice [88]. miR-195 promotes apoptosis in colorectal cancer cell lines by targeting antiapoptotic Bcl-2 [89]. Bcl-X(L) is regulated by miR-491 in CRC cells, and it suggests a therapeutic potential of miRNAs for treating CRC by targeting Bcl-X(L) [90]. miR-491-5p is differently expressed between DM and DKD patients [91]. miR-7 was significantly elevated in the type 2 DM patients and the type 2 DM-associated microvascular complications patients when compared with the controls [92]. Pancreatic β-cell failure underlay the progression of all forms of DM, and miR-7 acted as a brake on adult β-cell proliferation [93]. miR-7 was downregulated in CRC cell lines, and targeted the 3′ untranslated region of XRCC2 [94]. miR-148a suppressed the expression of Bcl-2 at the posttranscriptional level that leaded to activation of an intrinsic mitochondrial pathway and tumor apoptosis in CRC [95]. miR-148a-3p was dysregulated in type 1 DM patients compared to controls [51]. miR-148a-3p was associated with glucose levels and HbA1c levels [61]. An increased expression of miR-148a was observed in sera of type 1 DM patients compared with non-diabetic subjects [96].

miR-708 was significantly downregulated in CRC tissues and cell lines by targeting ZEB1 through AKT/mTOR signaling pathway [97]. miR-708 was identified as the most upregulated miRNA in islets cultured at low glucose concentrations [98]. Neuronatin might be a potential glucose-regulated target of miR-708 and miR-708 overexpression impaired GSIS, which was recovered by Neuronatin overexpression [98]. miR-182 might be a potential target for the treatment of diabetic sensory nerve regeneration because it was a key regulator in diabetic corneal nerve regeneration through targeting NOX4 [99]. Loss of miR-182 led to muscle fiber-type switching and impaired glucose metabolism [100]. miR-182 is related to insulin resistance by modulating FOXO1 and PI3K/AKT cascade [101]. An increased miR-182 expression may suppress the apoptotic pathway, promote cell proliferation, and confer aggressive traits on CRC cells [102]. miR-34a promotes apoptosis in the CRC cell line by targeting SIRT1 [103]. The expression levels of miR-34a were elevated in peripheral blood mononuclear cell samples from patients with type 2 DM [67]. The level of miR-34a-5p decreased in peripheral blood samples of type 2 DM patients compared with controls [104]. Ectopic expression of miR-133b inhibited CRC cell proliferation and caused cell cycle arrest in the G1 phase [105]. miR-133b is differently expressed between DM and DKD patients [91]. The expression levels of miR-133b were markedly depressed in the diabetic cardiomyocytes [106]. miR-145-5p was associated with survival for colon cancer patients [43]. Lentivirus-mediated miR-145 overexpression inhibited macrophage infiltration and improved glucose metabolism in db/db mice [107]. A study was performed to evaluate phenotype and function in vascular smooth muscle cells (SMC) cultured from non-diabetic and type 2 DM patients. Aberrant expression of miR-143/145 induced a distinct saphenous vein SMC phenotype in patients with type 2 DM [108]. miR-143 significantly reduces human colon cancer cell xenograft growth in vivo [109]. miR-143 impairs the insulin-AKT pathway, resulting in insulin tolerance and progression to type 2 DM [86].

miR-342 was dysregulated in type 1 DM patients compared to controls [51]. miR-342 reconstitution resulted in a marked increase in apoptosis in CRC cells [110]. Overexpression of miR-26b led to the significant suppression of the cell growth, and the inhibition of CRC growth in vivo [109]. miR-26b-5p was significantly different between ectosomes obtained from patients with type 2 DM and those obtained from healthy controls [111]. miR-26b-5p was found significantly downregulated following metformin treatments in patients with type 2 DM [112]. MiR-26b accelerated the progression of gestational DM by inhibiting the PI3K/Akt signaling pathway [113]. Re-expression of APC causes apoptosis in colon cancer by downregulating miR-135b [114]. Both in vitro and in vivo, the expression of miR-135b decreased in retinal cells under hyperglycemia exposure and increased in the DM retina [115]. miR-22 inhibited autophagy and promoted apoptosis to increase the sensitivity of CRC cells to 5-FU treatment both in vitro and in vivo [116]. The expression of miR-22 was increased in type 1 DM patients compared to the controls [117]. miR-22-3p antagonism improved glucose tolerance and insulin sensitivity [118]. miR-532-3p was downregulated both in colorectal adenoma and CRC [119]. DM leads to the downregulation of miR-532-3p expression in the skeletal muscle of male rats [120]. miR-532-3p was highly upregulated in male DM rats [121]. The expression level of miR-20a was associated with tumor necrosis factor-related apoptosis-inducing ligand (TRAIL) in CRC [122]. miR-20a-5p was significantly decreased in women with gestational DM compared with controls [123]. miR-20a was up-regulated in type 2 DM patients with non-alcoholic fatty liver disease (NAFLD) complicated compared to those without NAFLD [124].

PPARα was shown to be downregulated in the diabetic retina, which contributes to the pathogenesis of DR [125]. miR-21 targeted PPARα by inhibiting its mRNA translation and knockout of miR-21 prevented the decrease of PPARα and reduced cell apoptosis in the retina of db/db mice [126]. DKD patients with type 2 DM had higher urinary exosomal levels of miR-21-5p compared with type 2 DM patients with normal renal function [127]. Serum levels of miR-21-5p were increased in type 1 DM patients [96]. miR-21 may play an important role in the 5-FU resistance of colon cancer cells [128]. The down-regulation of the mismatch repair mutator gene associated with miR-21 overexpression may be an important indicator of therapeutic efficacy in CRC [129]. Anti-miR-21 mimics RhoB expression in inhibiting cell growth and invasion and inducing apoptosis of CRC cells [130].

A significant decrease in serum miR-17-3p in each of 30 nonproliferative diabetic retinopathy patients and 20 proliferative diabetic retinopathy patients when compared with healthy controls [131]. miR-17-5p can target and affect mitogen-activated protein kinases (MAPK) protein levels under high glucose conditions [122]. miR-17-5p had direct associations with BIRC5 for all CRC and increased expression of miR-17-5p, in carcinoma tissue improved survival [43]. Patients whose CRC tumors had high miR-17-5p expression had shorter overall survival rates but showed a better response to adjuvant chemotherapy than patients whose tumors had low miR-17-5p expression [132]. Insulin autoantibodies were negatively associated to miR-10b-5p [58]. miR-10b targeted components of the insulin signaling pathway [133]. Vitro studies showed the overexpression of miR-10b led to chemoresistance in CRC cells to 5-FU [134]. Anti-miR-196b increased apoptosis in CRC cell lines by upregulating FAS expression [135]. miR-196b-5p expression increased in serum and kidney of patients with DKD and miR-196b-5p-enriched extracellular vesicles mediated aldosterone-induced renal fibrosis in mice with DM [136].

The references for these common miRNA biomarkers of both diseases are presented in Table 1. In addition to these common miRNA biomarkers, DM and CRC have shared other miRNA biomarkers that we did not list in this review. It has been known that there is a strong relationship between these two diseases from the literature. This review provides their common miRNA biomarkers that may help gain a better understanding of the linking mechanism of DM and CRC.

## 4. Discussion

DM is caused by abnormalities of both insulin and glucose, and both are related to cancer cell proliferation. In addition to the common genetic factors such as miRNA, other common risk factors may contribute to the occurrence of both diseases such as obesity, sedentary behavior, western diet, and metabolic syndrome. An increased risk was observed for DM patients being obese for a total duration of 4 years or more [163]. The occurrence of obesity measured based on body mass index (BMI) in the colorectal adenoma positive patient group was significantly higher than the control group [164]. A sedentary lifestyle, obesity, and a Westernized diet have been implicated in the etiology of both CRC and non-insulin-dependent DM [165]. Sedentary behavior is associated with an increased risk of colon cancer and reducing sedentary behavior is potentially important for the prevention of CRC [166]. Risk factors such as sedentary lifestyle, obesity, Western diet, and metabolic syndrome are common in both type 2 DM and CRC [167]. These common risk factors for both DM and CRC are given in Figure 1.

Another evidence to link DM and CRC through the role of miRNAs can be discussed from the function of mitochondria. Mitochondria are the key regulator of glucose-stimulated insulin secretion in the pancreatic β-cells. Most of the adenosine triphosphate (ATP) synthesized during glucose metabolism is produced in the mitochondria. Mitochondrial dysfunction was suggested to play a key role in the pathophysiology of DM [168]. Some miRNAs were discovered to localize in mitochondria that are named as mitomiRs whether transcribed from the nuclear or the mitochondrial genome [169]. The identification of miRNAs in the mitochondria raised researchers’ interest to investigate the biological functions of mitomiRs. mitomiRs were discovered to influence various metabolic pathways such as tricarboxylic acid, lipid metabolism, and amino acid metabolism. These mitochondrial metabolic pathways are actively involved in energy metabolism during type 2 DM [170]. Glucose metabolism in human cells can be divided into two parts, which are oxidative phosphorylation (OXPHOS) and glycolysis in the cytosol in mitochondria [41]. Mitochondria in tumor cells are also responsible for the maintenance of cancer proliferation. miRNAs play a potential role in CRC cell metabolism related to mitochondria. Overexpression of miR-23a in CRC cells promoted the activation of pyruvate dehydrogenase (PDH) involved in OXPHOS to generate sufficient ATP for tumor cell proliferation [171]. miR-210 was shown to suppress mitochondrial respiration in CRC cells under the hypoxic condition [172]. miR-27a facilitates mitochondrial activity and glycolysis as well as promoting drug resistance in CRC cells [173].

Since both diseases are related, there have been studies investigating common medicine for both diseases. Metformin, an agent used in DM therapy, can increase insulin sensitivity. This fact suggested that metformin might have cancer growth inhibition potential and might be used as an anti-CRC agent. In a study of nearly 5000 patients with CRC and DM, it showed that the use of metformin was associated with improved survival relative to patients treated with other therapies for their DM [174]. Another study showed that CRC patients with DM, excluding those taking metformin, might have a worse CRC prognosis. The use of metformin, especially in the stage IV CRC population, might have a lower risk of dying [175]. However, in postmenopausal women with CRC and DM, no statistically significant difference was shown in CRC-specific survival in those who used metformin compared to non-users [176].

## 5. Conclusions

In this study, we review miRNAs that contribute to both DM and CRC. There are more common miRNA biomarkers than the miRNAs discussed in this paper for DM and CRC. Investigating these common miRNAs may shed a light on how these two diseases are correlated. The relationship between DM and CRC has been discussed in many cohort studies. It is important to explore this issue based on the molecular mechanism as well as clinical data analysis. Since both diseases share common risk factors, more understanding of the link between these two diseases can help the prevention of these diseases.

## Figures and Tables

**Figure 1 biomedicines-08-00530-f001:**
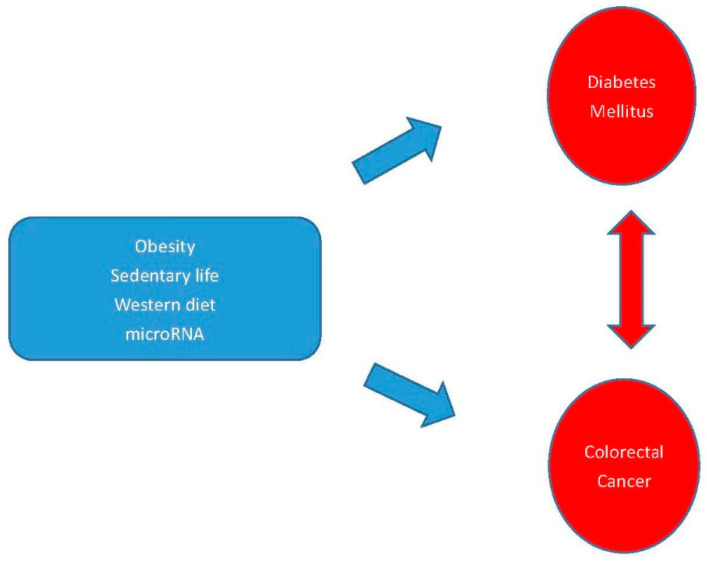
Some common risk factors and genetic factors of DM and CRC. The blue arrows show the risk factors cause to DM and CRC and the red arrow shows there is a relationship between DM and CRC.

**Table 1 biomedicines-08-00530-t001:** miRNAs related to Diabetes and Colorectal Cancer.

miRNA	Diabetes Reference	Colorectal Cancer Reference
miR-92a	[41,42]	[43,137]
miR-766	[44,45]	[46]
miR-21	[96,112,126,127,138,139]	[43,128,129,130,140,141,142]
miR-96	[48,49]	[47,140]
miR-17	[122,131]	[43,132]
miR-100	[50,51]	[52]
miR-365	[54,55,56,143]	[53,141]
miR-378	[57,58,144,145]	[34]
miR-18a	[60,61,143,146]	[59,147]
miR-125a	[63,64]	[62]
miR-125b	[66,67,68,148]	[65]
miR-10b	[58,133]	[134]
miR-200c	[55,70]	[69]
miR-206	[73,74,149]	[71,72,150]
miR-210	[51,76,151]	[75]
miR-23a	[78,79]	[77,152]
miR-129	[80]	[81]
miR-218	[82,83,84,85]	[86,153]
miR-195	[3,87,88]	[43,89]
miR-491	[91,154]	[90]
miR-7	[92,93]	[94]
miR-148a	[51,61,96]	[95]
miR-708	[54,98,155]	[97]
miR-182	[99,100,101]	[102]
miR-34a	[67,104]	[103,156]
miR-133b	[91,106]	[105]
miR-145	[107,108,157]	[43,105]
miR-143	[86,108,158]	[109,159]
miR-342	[51]	[110,160]
miR-26b	[111,112,113]	[109]
miR-135b	[115]	[114]
miR-196b	[136]	[43,135]
miR-22	[117,118,161]	[116]
miR-532	[120,121,162]	[119]
miR-20a	[123,124]	[122]

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
