# Peer review of "MicroRNA, Diabetes Mellitus and Colorectal Cancer"

_biomedicines, 2020, doi:10.3390/biomedicines8120530_

Round 1

Reviewer 1 Report

Comments to Authors:

Wang H  described diabetes and colorectal cancer pathology and the microRNA role as a biomarker in this review. The article required extensive revision and adhere to the journal style to write review articles. 

Major comments:

1. Abstract: The abstract was missing to link between miRNA and diabetes & CRC. The authors should emphasize the miRNA role in CRC & DM?

2.Introduction: "Diabetes mellitus (DM), known as Diabetes," poor write up in first sentence.The lines 23-24 are used the same as such in abstract.

3. In the "introduction" the authors describe much detail about diabetes(3 paragraphs). Instead they need much focus on the DM CRC link, how miRNA plays/acts as biomarkers in this context may be interesting to know the readers???

4. Discussion: This section write up was very poor. The author needs to revise this section to discuss more points.

5. This article needs diagrammatic illustration to explain colorectal cancer and diabetes link to understand the pathobiology?

Author Response

Thank you very much for the valuable comments. I have prepared a revised version incorporating the comments. The point-by-point responses are as follows. The revised parts are marked using the Microsoft Word "Track Changes" function in this revision.

  1. Abstract: The abstract was missing to link between miRNA and diabetes & CRC. The authors should emphasize the miRNA role in CRC & DM?

Response: In the abstract, we briefly illustrate some miRNA functions in CRC and DM.

This part was revised as follows.

“miRNAs can inhibit cell proliferation and induce apoptosis in CRC cells. miRNAs also can improve glucose tolerance and insulin sensitivity. Therefore, investigating the common miRNA biomarkers of both DM and CRC can shed a light on how these two diseases are correlated and more understanding of the link between these two diseases can help the prevention of both DM and CRC.”

2.Introduction: "Diabetes mellitus (DM), known as Diabetes," poor write up in first sentence.The lines 23-24 are used the same as such in abstract.

Response: "Diabetes mellitus (DM), known as Diabetes," was revised to “Diabetes mellitus (DM) referred to simply as diabetes”.

Lines 23-24. This sentence was revised to “Diabetes mellitus (DM) referred to simply as diabetes, is a metabolic disease that is due to either the pancreas not producing enough insulin, or the cells of the body not responding properly to insulin.”

  1. In the "introduction" the authors describe much detail about diabetes(3 paragraphs). Instead they need much focus on the DM CRC link, how miRNA plays/acts as biomarkers in this context may be interesting to know the readers???

Response: To much more focus on the DM and CRC link, I moved the first paragraph in Discussion section of the previous version to the end of Introduction section. This paragraph is about the clinical cohort studies that links DM and CRC. In addition, in Discussion section, I include a paragraph to provide another evidence to link DM and CRC through the role of miRNAs from the function of mitochondria.

A paragraph providing another evidence to link DM and CRC through the role of miRNAs from the function of mitochondria was added in the discussion section.

“Another evidence to link DM and CRC through the role of miRNAs can be discussed from the function of mitochondria. Mitochondria are the key regulator of glucose‐stimulated insulin secretion in the pancreatic β‐cells. Most of the adenosine triphosphate (ATP) synthesized during glucose metabolism is produced in the mitochondria. Mitochondrial dysfunction was suggested to play a key role in the pathophysiology of DM [170]. Some miRNAs were discovered to localize in mitochondria that are named as mitomiRs whether transcribed from the nuclear or the mitochondrial genome [171]. The identification of miRNAs in the mitochondria raised researchers’ interest to investigate the biological functions of mitomiRs. mitomiRs were discovered to influence various metabolic pathways such as tricarboxylic acid, lipid metabolism, and amino acid metabolism. These mitochondrial metabolic pathways are actively involved in energy metabolism during type 2 DM [172]. Glucose metabolism in human cells can be divided into two parts, which are oxidative phosphorylation (OXPHOS) and glycolysis in the cytosol in mitochondria [42]. Mitochondria in tumor cells are also responsible for the maintenance of cancer proliferation. miRNAs play a potential role in CRC cell metabolism related to mitochondria. Overexpression of miR-23a in CRC cells promoted the activation of pyruvate dehydrogenase (PDH) involved in OXPHOS to generate sufficient ATP for tumor cell proliferation [173]. miR-210 was shown to suppress mitochondrial respiration in CRC cells under the hypoxic condition [174]. miR-27a facilitates the mitochondrial activity and glycolysis as well as promoting drug resistance in CRC cells [175].”

  1. Discussion: This section write up was very poor. The author needs to revise this section to discuss more points.

Response: I added two paragraphs in the discussion section. The first paragraph discusses the common risk factors of DM and CMC. The second paragraph provides another evidence to link DM and CRC through the role of miRNAs from the function of mitochondria.

The first paragraph added in the discussion section:

“Diabetes is caused by abnormalities of both insulin and glucose, and both are related to cancer cell proliferation. In addition to the common genetic factors such as miRNA, other common risk factors may contribute to the occurrence of both diseases such as obesity, sedentary behavior, western diet, and metabolic syndrome. An increased risk was observed for diabetic patients who suffered from obesity for a total duration of 4 years or more [166]. The occurrence of obesity measured based on body mass index (BMI) in the colorectal adenoma positive patient group was significantly higher than the control group [167]. A sedentary lifestyle, obesity, and a Westernized diet have been implicated in the etiology of both CRC and non-insulin-dependent DM [168]. Sedentary behavior is associated with an increased risk of colon cancer and reducing sedentary behavior is potentially important for the prevention of CRC [169]. Risk factors such as sedentary lifestyle, obesity, Western diet, and metabolic syndrome are common in both type 2 DM and CRC [170]. These common risk factors for both DM and CRC are given in Figure 1.”

The second paragraph added in the discussion section:

“Another evidence to link DM and CRC through the role of miRNAs can be discussed from the function of mitochondria. Mitochondria are the key regulator of glucose‐stimulated insulin secretion in the pancreatic β‐cells. Most of the adenosine triphosphate (ATP) synthesized during glucose metabolism is produced in the mitochondria. Mitochondrial dysfunction was suggested to play a key role in the pathophysiology of DM [171]. Some miRNAs were discovered to localize in mitochondria that are named as mitomiRs whether transcribed from the nuclear or the mitochondrial genome [172]. The identification of miRNAs in the mitochondria raised researchers’ interest to investigate the biological functions of mitomiRs. mitomiRs were discovered to influence various metabolic pathways such as tricarboxylic acid, lipid metabolism, and amino acid metabolism. These mitochondrial metabolic pathways are actively involved in energy metabolism during type 2 DM [173]. Glucose metabolism in human cells can be divided into two parts, which are oxidative phosphorylation (OXPHOS) and glycolysis in the cytosol in mitochondria [42]. Mitochondria in tumor cells is also responsible for the maintenance of cancer proliferation. miRNAs play a potential role in CRC cell metabolism related to mitochondria. Overexpression of miR-23a in CRC cells promoted the activation of pyruvate dehydrogenase (PDH) involved in OXPHOS to generate sufficient ATP for tumor cell proliferation [174]. miR-210 was shown to suppress mitochondrial respiration in CRC cells under the hypoxic condition [175]. miR-27a facilitates the mitochondrial activity and glycolysis as well as promoting drug resistance in CRC cells [176].”

  1. This article needs diagrammatic illustration to explain colorectal cancer and diabetes link to understand the pathobiology?

Response: The pathobiology to link DM and CRC is related to the abnormalities of both insulin and glucose. The common risk factor of both diseases may cause abnormalities of insulin and glucose. Therefore, I include a paragraph in the discussion section (the first paragraph) to discuss some common risk factors of DM and CRC. A figure (Figure 1) was included to illustrate some common risk factors of DM and CRC.

Reviewer 2 Report

The author reviewed the relationship between DM and CRC based on their common miRNA biomarkers.

Concerning the manuscript goal, I think it is achieved.

However, given the importance of mitochondria, especially in DM, probably a more ellaborated discussion on this particular topic could be present, also given the fact that some studies on miRNAs/mitochondria, and its importance in cancer, have been put out (e.g. 10.1007/978-3-319-22671-2_8).

Line 302 Diabetes mellitus is written ans MD (should be DM).

Author Response

Thank you very much for the valuable comments. I have prepared a revised version incorporating the comments. The point-by-point responses are as follows. The revised parts are marked using the Microsoft Word "Track Changes" function in this revision.

1. Concerning the manuscript goal, I think it is achieved. However, given the importance of mitochondria, especially in DM, probably a more ellaborated discussion on this particular topic could be present, also given the fact that some studies on miRNAs/mitochondria, and its importance in cancer, have been put out (e.g. 10.1007/978-3-319-22671-2_8).

Response: Thank you for your comments. I included a paragraph in the discussion to link DM and CRC through the role of miRNAs from the function of mitochondria. The reference (10.1007/978-3-319-22671-2_8) was cited in this paragraph.

“Another evidence to link DM and CRC through the role of miRNAs can be discussed from the function of mitochondria. Mitochondria are the key regulator of glucose‐stimulated insulin secretion in the pancreatic β‐cells. Most of the adenosine triphosphate (ATP) synthesized during glucose metabolism is produced in the mitochondria. Mitochondrial dysfunction was suggested to play a key role in the pathophysiology of DM [171]. Some miRNAs were discovered to localize in mitochondria that are named as mitomiRs whether transcribed from the nuclear or the mitochondrial genome [172]. The identification of miRNAs in the mitochondria raised researchers’ interest to investigate the biological functions of mitomiRs. mitomiRs were discovered to influence various metabolic pathways such as tricarboxylic acid, lipid metabolism, and amino acid metabolism. These mitochondrial metabolic pathways are actively involved in energy metabolism during type 2 DM [173]. Glucose metabolism in human cells can be divided into two parts, which are oxidative phosphorylation (OXPHOS) and glycolysis in the cytosol in mitochondria [42]. Mitochondria in tumor cells are also responsible for the maintenance of cancer proliferation. miRNAs play a potential role in CRC cell metabolism related to mitochondria. Overexpression of miR-23a in CRC cells promoted the activation of pyruvate dehydrogenase (PDH) involved in OXPHOS to generate sufficient ATP for tumor cell proliferation [174]. miR-210 was shown to suppress mitochondrial respiration in CRC cells under the hypoxic condition [175]. miR-27a facilitates the mitochondrial activity and glycolysis as well as promoting drug resistance in CRC cells [176].”

In addition, a paragraph about the common risk factors of DM and CMC was added in the discussion section.

 “Diabetes is caused by abnormalities of both insulin and glucose, and both are related to cancer cell proliferation. In addition to the common genetic factors such as miRNA, other common risk factors may contribute to the occurrence of both diseases such as obesity, sedentary behavior, western diet, and metabolic syndrome. An increased risk was observed for diabetic patients who suffered from obesity for a total duration of 4 years or more [166]. The occurrence of obesity measured based on body mass index (BMI) in the colorectal adenoma positive patient group was significantly higher than the control group [167]. A sedentary lifestyle, obesity, and a Westernized diet have been implicated in the etiology of both CRC and non-insulin-dependent DM [168]. Sedentary behavior is associated with an increased risk of colon cancer and reducing sedentary behavior is potentially important for the prevention of CRC [169]. Risk factors such as sedentary lifestyle, obesity, Western diet, and metabolic syndrome are common in both type 2 DM and CRC [170]. These common risk factors for both DM and CRC are given in Figure 1.”

2. Line 302 Diabetes mellitus is written and MD (should be DM).

Response: Lines 324-325. “MD” was revised to “DM”.

Round 2

Reviewer 1 Report

The manuscript may be acceptable